# Online repositories of photographs and videos provide insights into the evolution of skilled hindlimb movements in birds

Cristián Gutiérrez-Ibáñez [1,5✉], Clara Amaral-Peçanha[2,5], Andrew N. Iwaniuk [3], Douglas R. Wylie [1] & Jerome Baron[2,4]

The ability to manipulate objects with limbs has evolved repeatedly among land tetrapods. Several selective forces have been proposed to explain the emergence of forelimb manipulation, however, work has been largely restricted to mammals, which prevents the testing of evolutionary hypotheses in a comprehensive evolutionary framework. In birds, forelimbs have gained the exclusive function of flight, with grasping transferred predominantly to the beak. In some birds, the feet are also used in manipulative tasks and appear to share some features with manual grasping and prehension in mammals, but this has not been systematically investigated. Here we use large online repositories of photographs and videos to quantify foot manipulative skills across a large sample of bird species (>1000 species). Our results show that a complex interaction between niche, diet and phylogeny drive the evolution of manipulative skills with the feet in birds. Furthermore, we provide strong support for the proposition that an arboreal niche is a key element in the evolution of manipulation in land vertebrates. Our systematic comparison of foot use in birds provides a solid base for understanding morphological and neural adaptations for foot use in birds, and for studying the convergent evolution of manipulative skills in birds and mammals.

[1] Department of Biological Sciences, University of Alberta, Edmonton, Canada. [2] Graduate Program in Physiology and Pharmacology Institute of Biological Sciences, Federal University of Minas Gerais, Belo Horizonte, Brazil. [3] Department of Neuroscience, Canadian Centre for Behavioural Neuroscience, University of Lethbridge, Lethbridge, Alta., Canada. [4] Department of Physiology and Biophysics, Institute of Biological Sciences, Federal University of Minas Gerais, Belo Horizonte, Brazil. [5] These authors contributed equally: Cristián Gutiérrez-Ibáñez, Clara Amaral-Peçanha. ✉email: cgutierr@ualberta.ca

The ability to grasp or manipulate objects with appendages has evolved repeatedly among land tetrapods[1]. Because grasping and manipulating objects are characteristic of humans and nonhuman primates, the neural basis of these behaviors, and their association with primate evolution, including the brain, have received extensive attention[1–4]. In the case of primates brain evolution, the development of skilled manipulation has been related to the evolution of specialized visual and motor circuits[2,5].Given the importance of skilled forelimb manipulation in a variety of behaviors[4], a significant question is what drives the evolution of grasping and manipulation in some, but not all, species. In mammals, several selective forces have been proposed to explain the emergence of forelimb manipulation, including arboreal locomotion, digging, and prey handling[1,4]. To better test this hypothesis, understanding the evolution of manipulation with the extremities in other vertebrates is needed, but this behavior has received relatively little attention outside the mammalian literature[1].

In birds, forelimbs have gained the almost exclusive function of flight, with grasping transferred predominantly to the beak[6,7]. However, the absence of a second extremity limits the ability to manipulate objects with the beak. Consequently, many birds have evolved the ability to grasp and manipulate objects with their feet[4,8], including hawks, owls and falcons, which use their feet to capture and hold prey[9,10], as well as parrots, mousebirds and many songbirds[8,11]. Given the diversity of clades in which pedal manipulation has evolved, birds represent a key comparison for understanding the evolutionary pathways by which pedal and manual dexterity have evolved in tetrapods. Unfortunately, there have been no systematic studies on the evolution of pedal dexterity in birds. A review of published reports by Sustaita et al.[4] suggests that arboreality predates manipulative foot use in birds, although this was based on only a limited species sample (around 150 species). A broader approach is required to establish if the development of manipulative foot use aligns with the evolution of arboreality in birds. In addition, the extent to which manipulation skills vary among and within different avian clades is unknown. For example, some parrots use their feet to bring food to their beak and coordinate beak and foot movements for extractive foraging[12], but this does not appear to be true of all parrots[13]. Whether similar manipulation skills have evolved outside parrots is equally unclear.

In mammals, several studies have used direct observations of animals in captivity to evaluate differences in manipulative skills across species[14–17]. While direct observations allow for a detailed study of manipulative skills, relatively few species can be examined this way, thus preventing the testing of hypotheses in a comprehensive evolutionary framework[1,18,19]. An alternative approach is to use data deposited in digital databases by citizen scientists[20]. This approach is particularly suitable for studies on birds, as a large (and growing) collection of pictures and videos are available: Macaulay Library at the Cornell Lab of Ornithology alone has >40 million pictures and videos of birds (https://macaulaylibrary.org/). Here, we use large online repositories of photographs and videos, as well as previous literature, to quantify foot manipulative skills of birds (Fig. 1a) and test several hypotheses about the selective pressures that give rise to skilled manipulation with the limbs in tetrapods.

## Results

Our citizen-science approach (Fig. 1a) allowed us to obtain and score 3725 individual media files of birds using their feet to manipulate objects (Dataset 1) from a variety of sources (Fig. 1b). The observations encompassed 1054 species (i.e., close to 10% of all bird species) belonging to 13 orders and 64 families (Dataset 2). For clades where we systematically searched for foot use in all species (see Methods), we found media of foot use behavior in 40 to 95% of species (Fig. 1c, d) in those clades. This large data set not only allowed us to assess if manipulation of objects with the feet was present, but also allowed us to quantify, in detail, (Fig. 1a, Table 1) the manipulative skills at the species or at the very least, genus level.

**Arboreality drives the evolution of foot use in birds**. First, we considered the absence or presence of foot use (for manipulation) at the family level to understand the origin of this behavioral trait in birds. The outer circle of dots in Fig. 2a shows the presence or absence of foot use, as well as families for which insufficient data were available (see Methods), in all 250 families of birds. We then performed an ancestral state reconstruction using a hidden Markov model. Our results show that the best supported model (Supplementary Fig. 1) is one where the transition from no foot use to foot use is indirect, through a "precursor" state, for instance, arboreality. Our results (Fig. 2a) show that the transition from an absence of foot use to this precursor occurred only once in the avian phylogeny at the base of the Telluraves, the clade that includes most of the small, arboreal neornithine birds[21], and that after the transition to this precursor, foot use emerged independently at least 20 times (Fig. 2a). This includes independent origins of foot use for each of the raptor clades (falcons, owls, and hawks)[9], mousebirds (Coliiformes), and parrots (Psittaciformes). We also found a single independent origin of foot use within the order Piciformes for a monophyletic clade comprising toucan-barbets (Semnornithidae), New-World barbets (Capitonidae), and toucans (Ramphastidae). Additionally, there were at least 14 independent gains of foot use within songbirds (Passeriformes). Most strikingly, our analysis recovers foot use as the most likely ancestral state of two large radiations of songbirds: the suborder Corvides[22] and the superfamily Sylvioidea (Fig. 2a[23],). Outside Telluraves, foot use is rare and has only evolved four times in <15 species. The most notable of these is the evolution of grasping and the ability to bring the foot and objects to the beak in eight species of swamphens that belong exclusively to the genus Porphyrio[24]. Additionally, foot use has evolved in several individual species nested within larger clades: the greater coucal (Centropus sinensis, Cuculiformes), Australian brush-turkey (Alectura lathami, Galliformes), and snowy sheathbill (Chionis albus, Charadriiformes).

**Differences in pedal manipulation skills and diet**. Not only has foot use evolved independently multiple times in birds, but there are also significant differences in the manipulative skill among clades (Phylogenetic Generalized Least Squares (PGLS), $F_{5:1020} = 3.09$ $p = <0.01$; Supplementary Table 2, Fig. 2b, c). Most foot-using songbirds and piciforms have relatively simple manipulative skills (Fig. 2b, Supplementary Fig. 1b), consisting primarily of holding or clasping against a surface, with only a few species capable of grasping. The three raptorial orders have higher manipulative skills, associated with the widespread ability to grasp objects and, in many cases, bring objects to the beak (see below). Finally, parrots had the highest scores for our skill index. This is driven by the capacity of most parrots to grasp and bring object to the beak while also rotating their foot to manipulate objects, which includes not only food items but also tools and others non-food objects[25,26].

Next, we wanted to test if the evolution of foot use is related to a particular diet. First, we looked at the percentage of species that use their foot within each diet category (Fig. 2d, foot use, dark blue bars; yellow bars, no evidence of foot use). We also included all species that belong to families that use their feet, but for which we did not have any observations of foot use (Fig. 2d, light blue bars). This procedure is likely to overestimate the number of species that

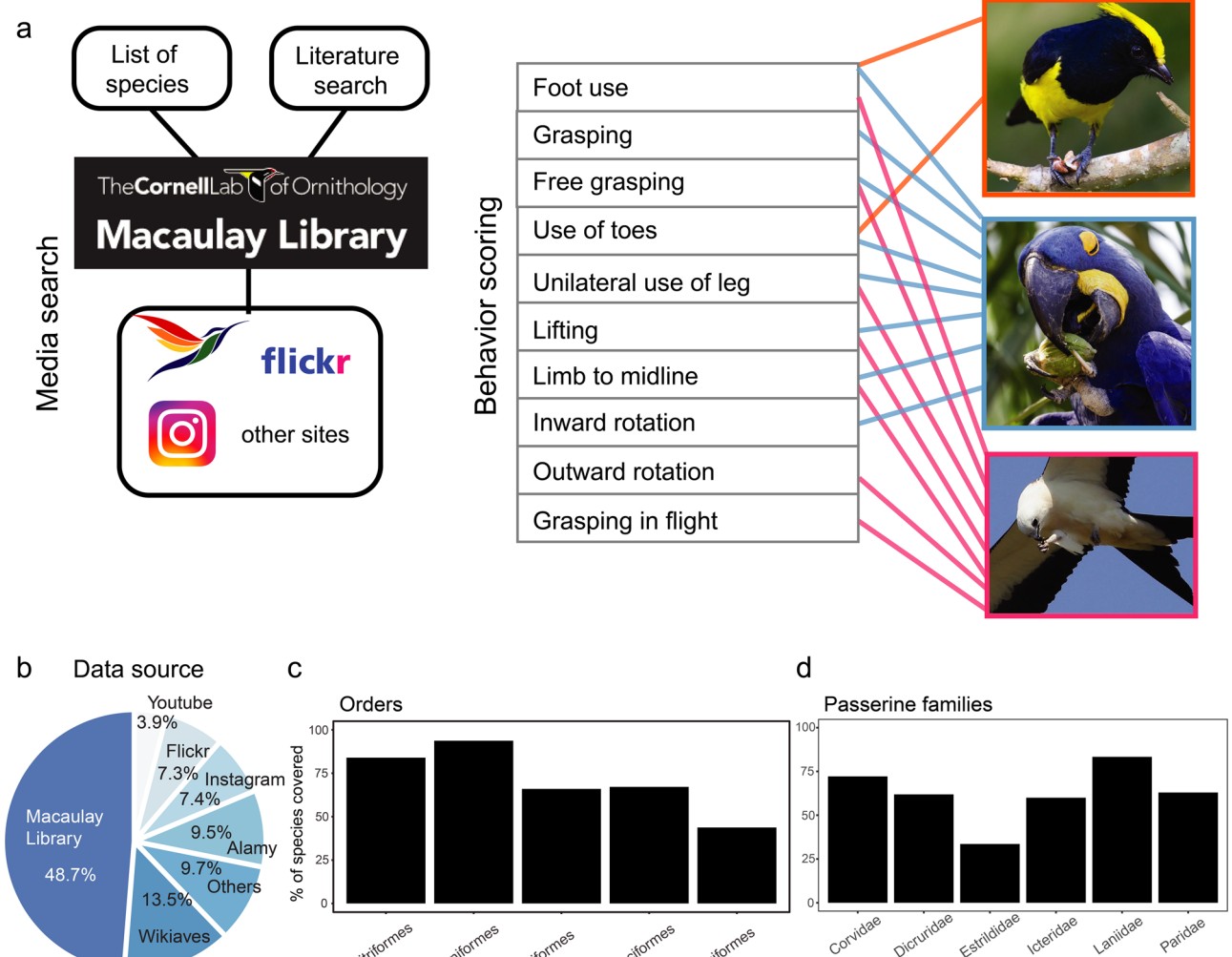

**Fig. 1 Online bird media repositories are an effective source of behavioral data. a** We combined exhaustive searches of media in clades known to use their feet with extensive literature searches to determine which clades use their feet to manipulate objects as well as compare manipulative skills among birds. Photographs, top to bottom: Sultan Tit (*Melanochlora sultanea*), Hyacinth Macaw (*Anodorhynchus hyacinthinus*), Swallow-tailed Kite (*Elanoides forficatus*). Photographer credits are listed in Supplementary Table 6. **b** Approximately 60% of media came from two citizen science bird media repositories (Macaulay Library and Wikiaves). **c, d** Species coverage for different orders and songbird families we searched systematically (all species). Coverage was at least 40% and as high as 95%.

**Table 1 Behavioral scoring for each media of bird using their feet to manipulate objects.**

| Behavior | Description | Score |
|---|---|---|
| Foot use | The bird uses its claw to hold or grasp an object | 0,1 |
| Grasping | Claw/toes closed around an object | 0,1 |
| Free grasping | Claw/toes closed around an object while object is not in contact with any surface. Other parts of the claw/leg can be in contact with the surface. | 0,1 |
| Free grasping in flight | Object grasped during flight | 0,1 |
| Use of toes | Object is held or grasped with only some toes | 0,1 |
| Use of one leg | Only one leg is used to hold or grasp | 0,1 |
| Foot to beak | Claw is lifted to the beak | 0,1 |
| Foot to midline | Claw is brought to the midline | 0,1 |
| Inward rotation | The claw is rotated inward | 0,1 |
| Outward rotation | The foot is rotated outward | 0,1 |

use their feet in some diet categories, but was included as our sampling was not uniform across birds. Although some diets seem to be associated with foot use (vertebrates, carrion), species with any diet can potentially manipulate objects with their feet. Because diet changes could have occurred after the origin of foot use, we next performed an ancestral state reconstruction of diet in a large sample of birds to assess the ancestral diet at the main nodes where foot use has evolved independently (Supplementary Table 3). Here we found that the origin of foot use is associated with at least four different diets: vertebrates (reconstructed for the ancestors of the

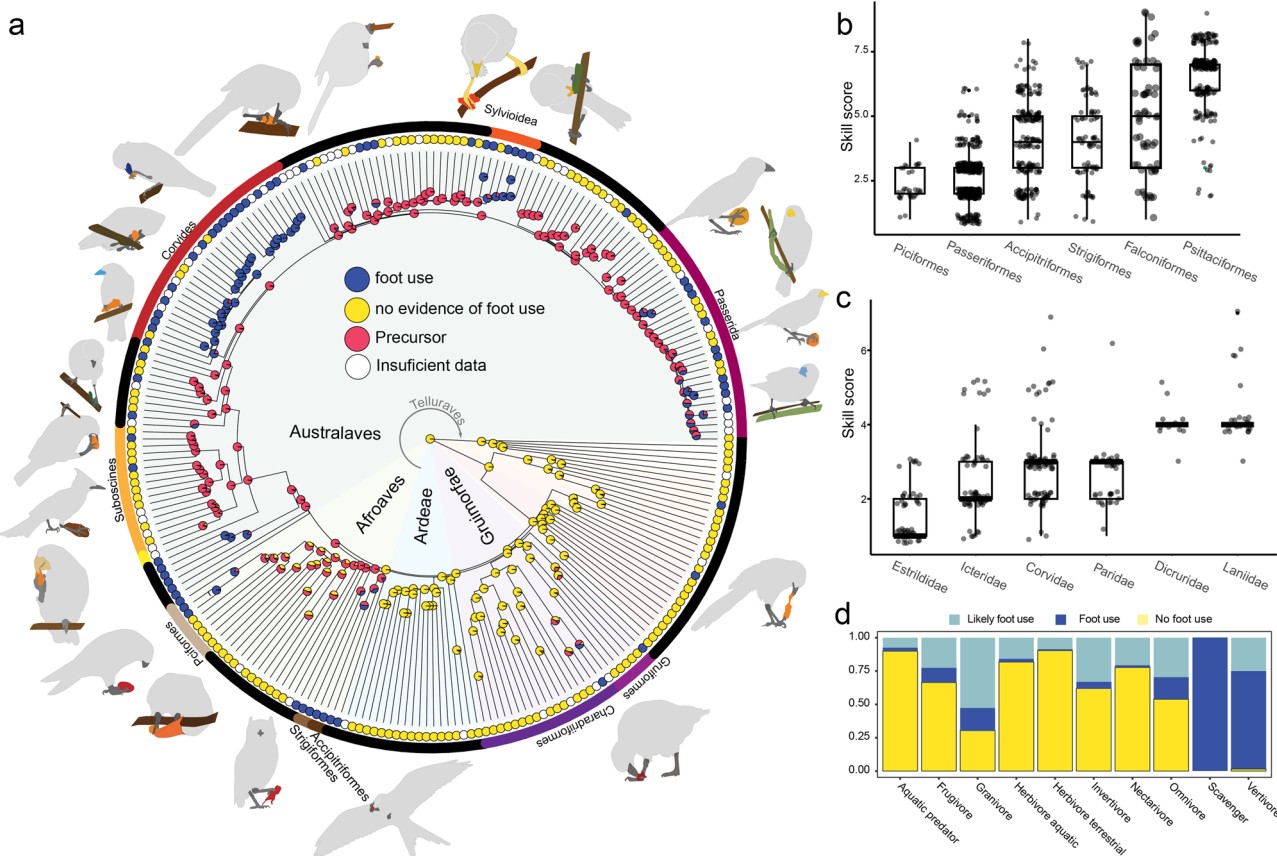

**Fig. 2 After the transition to arboreality, foot use has repeatedly evolved in different clades, and is associated with multiple diets. a** Ancestral state reconstruction with a precursor model shows that after a single evolution of a precursor state at the base of all core land birds (Telluraves), foot use has evolved repeatedly, at least 20 times. **b, c** Skill scores for different orders (**b**) and songbird families (**c**). The lower and upper hinges correspond to the first and third quartiles. Whisker extends no further than 1.5 the inter-quartile range, or distance between the first and third quartiles. **d** Percentage of species for each diet category where there is no evidence of foot use (yellow), evidence of foot use (dark blue) and likely foot use (light blue). Species are considered to belong to a particular diet category if >60% of the diet is of one type[72]. Likely foot use refers to species that belong to families where foot use was confirmed, but those species where not searched. In general, foot use is not associated with only one diet (see Supplementary Table 3).

three raptors orders), invertebrates (for the two large passerine radiations, Corvides and Sylvioidea), grains (parrots and at least two clades in the super order Passerida), and fruit (in South American barbets and toucans) (Supplementary Table 3, Supplementary Fig. 1a). These results indicate that the independent evolution of foot use in birds is not driven by a specific diet and that after the transition to arboreality, foot use can be co-opted to manipulate a variety of food items.

**Foot use in raptors**. Given that the three raptorial orders have evolved foot use independently but with similar diets and morphology[9], we wanted to see if they shared similar skills in foot use. Figure 3 shows genus-level phylogenies and behavioral character matrices for the three raptor orders. Genera for which insufficient data were available were omitted (see Methods). Not surprisingly, foot use is widespread in all three orders and is the ancestral condition for each of them. Falcons and owls share similar skills; most genera are capable of free grasping and lifting objects to their beaks (Fig. 3a–c). In contrast, within Accipitriformes (hawks, eagles, and allies) the ability to lift objects to the beak while perching is rare and only partially present in a few clades (Fig. 3d). Curiously, many more accipitriform genera bring objects to their beaks during flight (e.g., Fig. 3a, bottom panel; Fig. 3d). Additionally, we found a convergence in how New and Old World vultures use their feet. In New-World vultures (Cathartidae, purple in Fig. 3d) and one of the Old World vulture

clades (Aegypiinae[27], green in Fig. 3d), most species are incapable of grasping objects and only hold prey against a surface. In other words, these two vulture clades share a rudimentary manipulation of food with their feet. In Old World vultures, this implies a loss of grasping ability, since Old-World vultures are nested within Accipitriformes, where grasping is widespread and ancestral. Thus, a change in diet, in this case from vertivore to scavenger, leads to a change in foot use skill. Finally, to examine other traits that may affect differences in manipulative skills among raptorial orders in more detail, we also compared their diets and body mass (Fig. 3e, f). Falcons and owls are similar in both body mass and diet, and on average are significantly smaller (PGLS, $F_{3,518} = 86.9$, $p = <0.0001$, Supplementary Table 2) than accipitriforms. The smaller mass of falcon and owls is reflected in a much higher percentage of species within these groups that feed on invertebrates (Fig. 3f), which likely explains the higher skill indices of these groups compared to accipitriforms.

**Foot use varies among parrots**. In contrast to falcons and owls, where foot use skills are similar in all species, foot use skills vary greatly among parrots. Our citizen science approach allowed us to score foot use in >70% of all parrot species, allowing us to explore foot use variations in detail (Fig. 4a). An ancestral state reconstruction (Fig. 4b) shows that while foot use is ancestral among parrots, it has been lost or reduced at least seven times independently. This includes the loss of foot use in several smaller

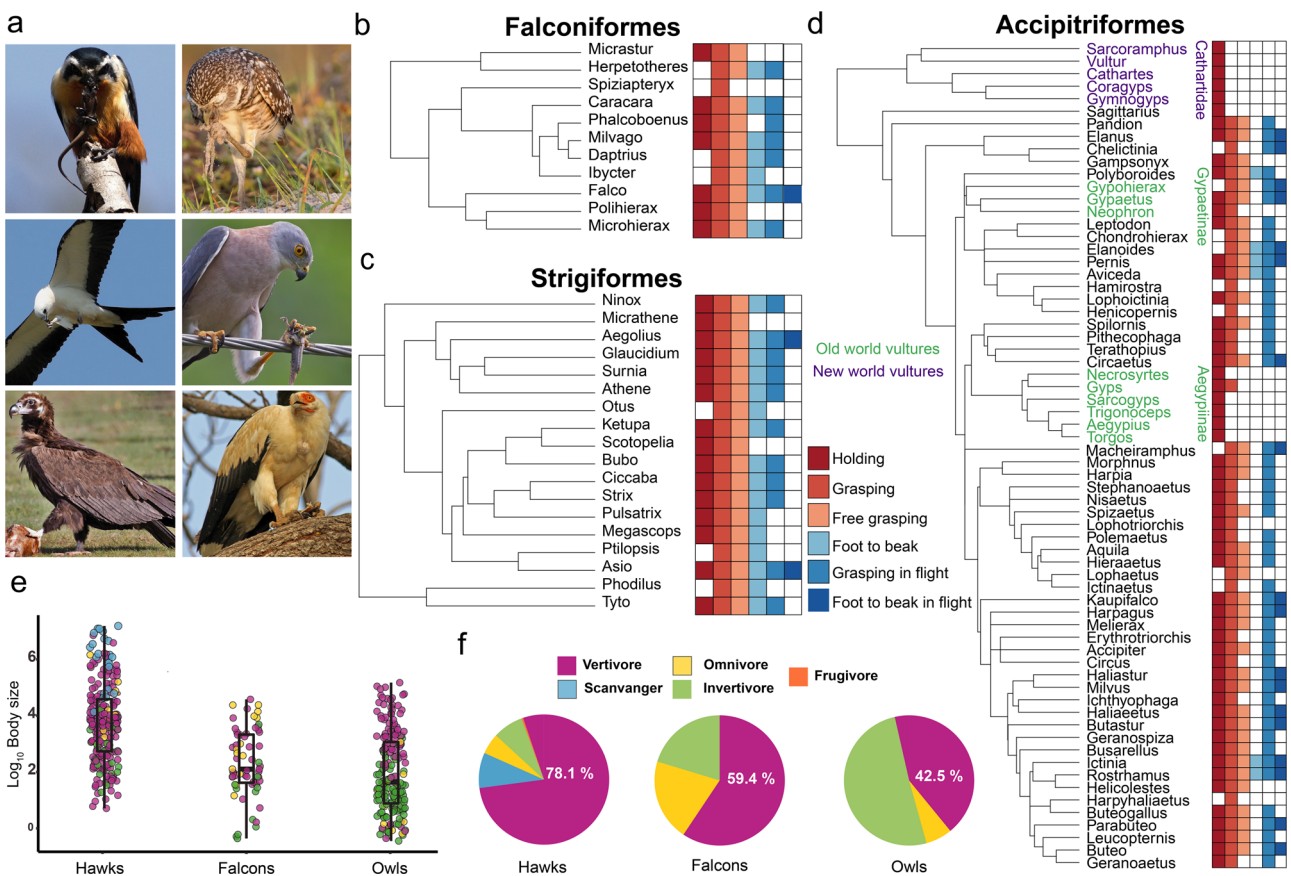

**Fig. 3 Variation in foot use between and within raptors. a** Examples of the three different raptor orders using their feet to manipulate objects. Top row, left to right: Collared Falconet (*Microhierax caerulescens*), Burrowing Owl (*Athene cunicularia*), Middle row, left to right: Swallow-tailed Kite (*Elanoides forficatus*), Fiji Goshawk (*Accipiter rufitorques*). Bottom row, left to right: Palm-nut Vulture (*Gypohierax angolensis*), Cinereous Vulture (*Aegypius monachus*). Photographer credits are listed in Supplementary Table 6. **b–d** Genus level phylogenies for falcons (**b**), owls (**c**) and hawks, eagles and New-World vultures (**d**) showing the different foot use behavior present for each genus. The key for the skilled foot use matrix is provided in (**c**), with colored squares reflecting the presence of each of the six behavioral elements. Falcons and owls have similar matrices in which the ability to grasp objects and bring them to the beak is widespread and likely ancestral. In contrast, the ability to bring the foot to the beak while perching (fourth column of the matrix) is only present in five genera of Accipitriformes, but the ability to bring the foot to the beak (sixth column) while flying is more widespread. Also shown is the convergent loss of grasping in New World vultures (in purple) and one of the Old World vulture clades (in green). **e** Body masses of the three orders of raptors. The color of each dot corresponds to the diet of each species as shown in **f**. The lower and upper hinges correspond to the first and third quartiles. Whisker extends no further than 1.5 the inter-quartile range, or distance between the first and third quartiles. **f** Percentage of each diet category in each raptor order. See methods for details on how dietary categories were assigned.

genera: *Forpus, Neophema, Neopsephotus, Touit, Melopsittacus, Cyclopsitta, Agapornis,* and *Micropsitta*. The lories and lorikeets (Loriini) had the greatest diversity in foot use; it is reduced or absent in several species (e.g., *Psitteuteles*), but other species (e.g., *Trichoglossus*) have manipulative skills similar to other parrots, including the ability to grasp and bring objects to the beak. Loss of foot use is not clearly associated with one diet or niche (Supplementary Fig. 2a-d). Several of the clades where foot use has been lost or reduced are largely granivorous and feed primarily on the ground (e.g., *Neophema, Agapornis,* and *Melopsittacus*), but some perching and frugivorous genera also lost foot use (for example, *Touit,* Supplementary Fig. 2).

Most parrots grasp objects with the two external toes and/or turn their leg inward when bringing objects to their beaks (Fig. 4a). Our ancestral state reconstruction recovers this behavior as the ancestral state for all parrots (Fig. 4b, c). However, two clades have independently evolved the ability to grasp objects with the inner toes and turn their legs outwards when bringing objects to their beak (Fig. 4). One of these independent changes to outward rotation is at the base of the tribe Androglossini (*Amazona* and *Pionus* species, as well as related genera[28]). The

second clade is Psittaculini (*Psittacula, Eclectus,* and related genera), although the racket-tails (genus *Prioniturus,* Fig. 4b, c) do not appear to rotate the foot outwards. As with loss of foot use, it is unclear whether the emergence of this new manipulative skill is associated with a particular diet, niche, or lifestyle (Supplementary Fig. 2).

**Variation of foot use in songbirds**. As mentioned above, we found repeated and independent evolution of foot use among passerines (at least 14 times, Fig. 2A), largely concentrated in oscines. Only two families of suboscines, Furnariidae and Oxyruncidae, exhibit foot use. In general, passerines have lower skill scores than orders in which foot use is common (Fig. 2b, c), which is reflected in their tendency to hold objects against the ground or perch but not grasping (Supplementary Fig. 3). The exceptions are families within the superorder Corvides, where the ability to freely grasp objects occurs in many families and the ability to bring objects to the beak has evolved in at least three different families: shrikes (Laniidae), drongos (Dricuridae) and vangas (Vanguidae) (Fig. 5a, b). Outside of this clade, although there are a few species capable of grasping objects while hanging

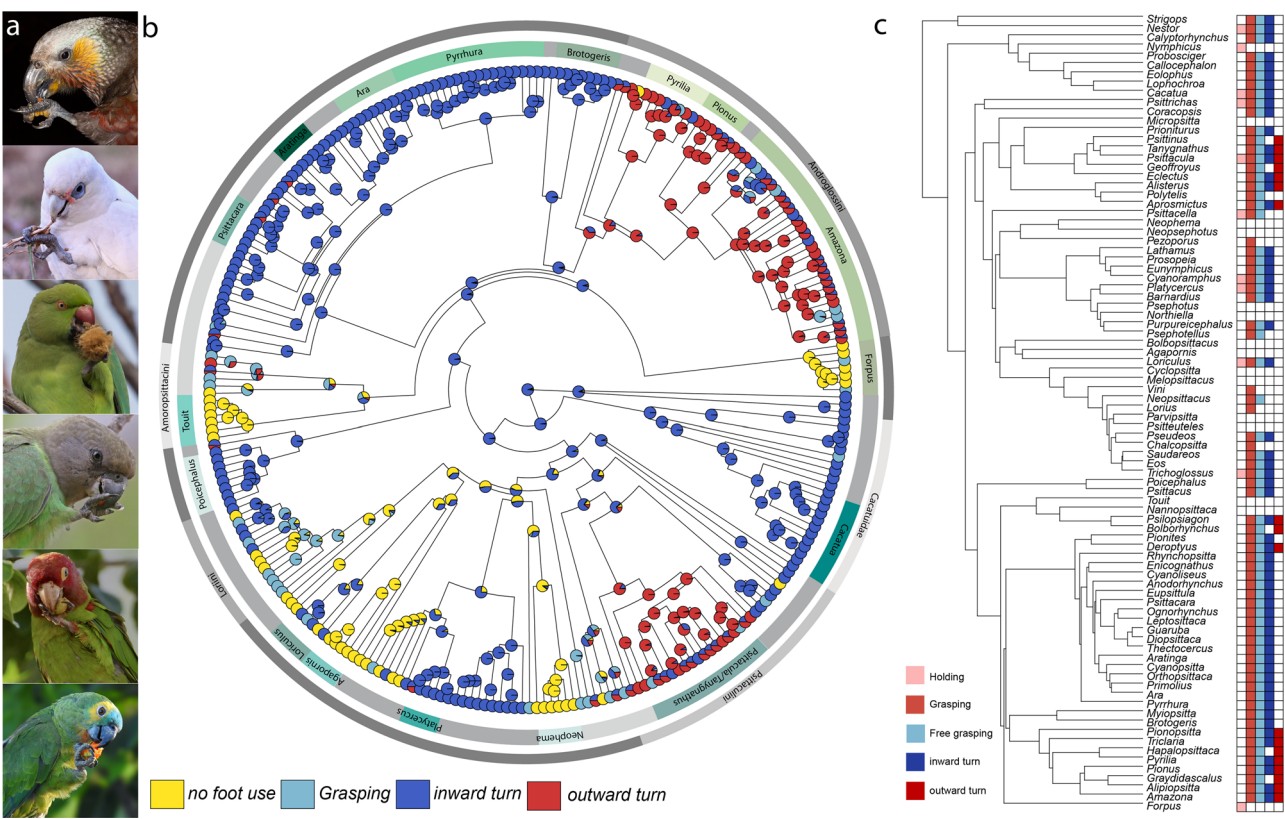

**Fig. 4 Variation in foot use among parrots shows several instances of convergent evolution. a** Examples of foot use among the main clades of parrots. From top to bottom: New Zealand Kaka (*Nestor meridionalis*), Red-masked Parakeet (*Psittacara erythrogenys*), Brown-headed Parrot (*Poicephalus cryptoxanthus*), Turquoise-fronted Parrot (*Amazona aestiva*), Little Corella (*Cacatua sanguinea*), Rose-ringed Parakeet (*Psittacula krameria*). Photographer credits are listed in Supplementary Table 6. **b** Ancestral state reconstruction of foot use in parrots shows that foot use is the ancestral state for all parrots, but this behavior has been lost at least five times. The ancestral state reconstruction also shows that an outward rotation of the foot when bringing the foot to the beak has evolved independently at least twice (dark red). **c** Genus level phylogeny of parrot shows in detail where foot use has been lost and where differences in foot use have emerged. Colored squares reflect the presence of each of five different behavioral elements.

from one foot (e.g., Aegithalidae, Remizidae, Paridae), no other species are capable of bringing an object to the beak while perching. Only one family, Callaeidae (New Zealand wattlebirds), is capable of free grasping.

To better understand what might drive differences in manipulation skill among passerines, we systematically searched for foot use in three closely related families within the superfamily Corvoidea: drongos (Dricuridae), shrikes (Laniidae) and crows, jays and magpies (Corvidae) (Fig. 5c–e). Most drongo and shrike species use their feet to grasp objects and are capable of free grasping and some drongos and shrikes can even bring objects to the beak with their foot (Fig. 5c, d). In contrast, grasping is rare in corvids: only one species appears to have the ability to free grasp (the yellow-billed chough, *Pyrrhocorax graculus*), and no corvids bring objects to their beaks with their feet (Fig. 5e). Corvidae differ from the other two families in that their diet is largely omnivorous, whereas drongos and shrikes are predators that feed on insects and small vertebrates (Fig. 5). Thus, the ability to manipulate objects with the feet is not associated with dietary breadth, but rather specific dietary types in songbirds.

## Discussion

Here we have taken advantage of the vast number of pictures and videos of birds stored in online citizen science repositories to study the evolution of a largely ignored behavior in birds: skilled pedal manipulation. Our results show that this approach can result in extensive coverage (up to 95% of species in some clades, Fig. 1b), and that is sensitive enough to detect small and

previously undescribed differences in behavior, such as the differences in manipulation among raptor clades (Fig. 3), the emergence of new manipulation skills among parrots (Fig. 4), and the loss of foot use in some parrots (Fig. 4, Supplementary Fig. 3). While some studies have already used this resource to study different aspects of bird biology,[e.g 29,30.] the scale at which we employed this approach (covering 10% of all birds) is unprecedented. Further, this is the first study outside of mammals to quantify and compare manipulative skills in a broad phylogenetic context and test which factors might drive the evolution of limb manipulation across vertebrates[e.g 14].

In mammals the evolution of manipulation with the forelimb is typically associated with arboreality, digging, and prey manipulation ([Reviewed in 4]). Here we show that the evolution of object manipulation with the feet in birds is a complex interaction of several putative selective forces. First, our results support the previous suggestion[4] that the evolution of foot use in birds is greatly facilitated by the transition to arboreality. Our ancestral state reconstruction shows that the most likely scenario for the evolution foot manipulation in birds is one where a transitional state is required, and that this state evolved once at the node that gave rise to the core landbirds, Telluraves. Given that this clade includes most of the small arboreal neornithine birds[21] and arboreality is the ancestral state of all Telluraves[31], we suggest that the transitional state required from the evolution of foot use is most likely an arboreal niche. Our ancestral state reconstruction of ancestral diet among the principal clades that have independently evolved foot use suggests that this behavior is related to

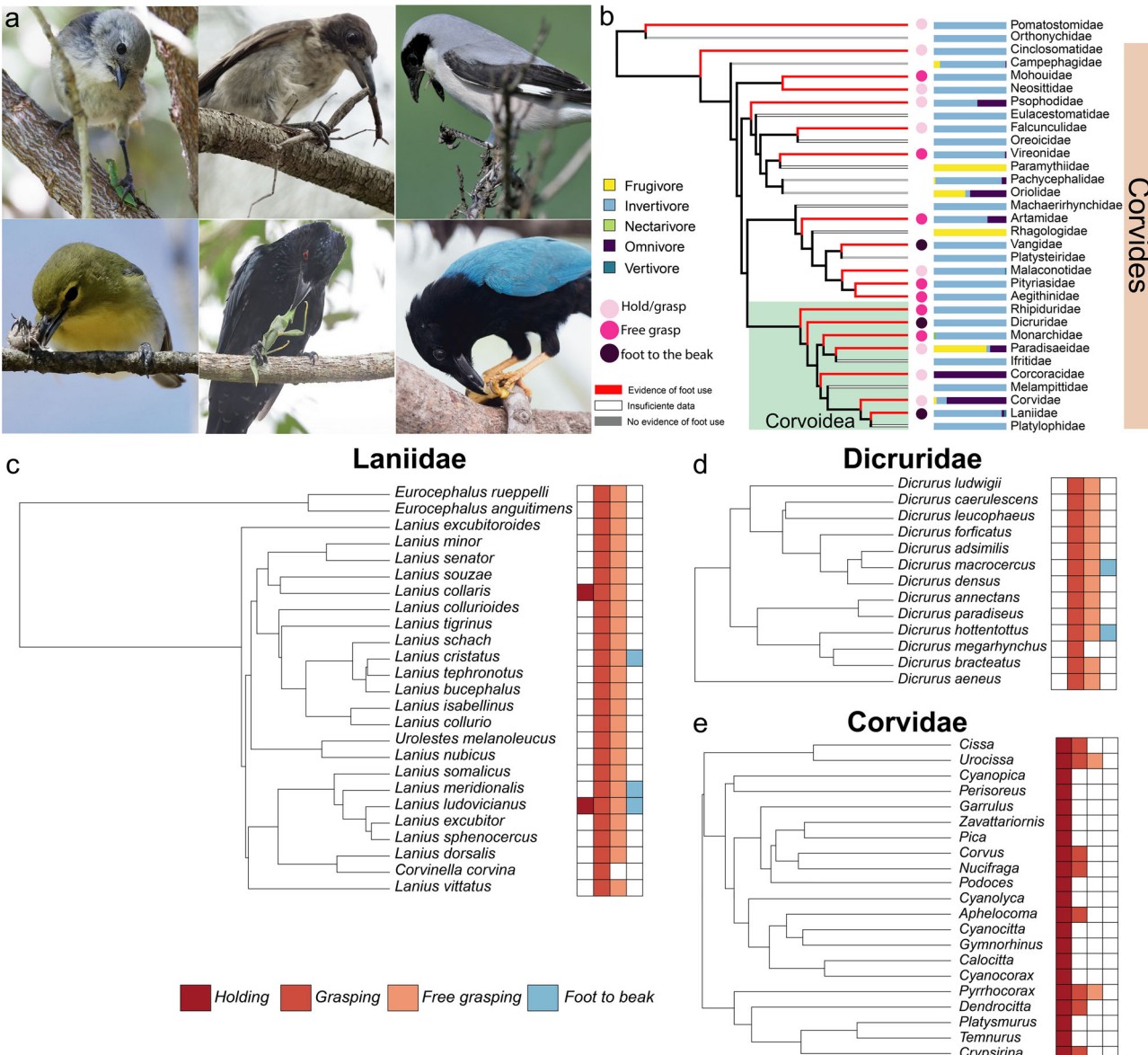

**Fig. 5 Changes in foot use are driven by diet changes among the suborder Corvides. a** Photographs showing examples of foot use among 6 of the 19 families within the superorder Corvides that use their feet to manipulate objects. Top row, from left to right: Whitehead (*Mohoua albicilla*), Pied Butcherbird (*Cracticus nigrogularis*), Lesser Gray Shrike (*Lanius minor*). Bottom row: Yellow-throated Vireo (*Vireo flavifrons*), Spangled Drongo (*Dicrurus bracteatus*), Yucatan jay (*Cyanocorax yucatanicus*). Photographer credits are listed in Supplementary Table 6. **b** Family level phylogeny of the suborder Corvides shows widespread foot use and that most of the Corvides families are insectivores, with only a few shifts to omnivory or frugivory. Bars represent the percentage of species for each within each diet category and the circles show the different manipulation skills of each family. **c, d** Show a character matrix for shrikes (Laniidae) and drongos (Dicruridae) adjacent to species level phylogenies. Both families are almost strictly invertivores and most species have the ability to free grasp and even bring object to the beak. In contrast, the character matrix at the genus level for crows and jays (**e**, Corvidae) shows that most species do not grasp.

different diets in different clades (Fig. 2, Supplementary Table 3). Thus, once birds evolved the morphological traits for perching, these traits were subsequently and repeatedly exapted for grasping and manipulating food items that included, but were not limited to, vertebrate and invertebrate prey. However, the relationship between foot use and diet is not straightforward. In the barbets and toucans, for example (Supplementary Fig. 1), frugivory is the ancestral state for a clade that includes three families that use their feet to manipulate objects, as well as two other families like the Asian (Megalaimidae) and African barbets (Lybiidae) that do not appear to use their feet to manipulate objects. Similarly, hornbills (Bucerotidae) have convergently evolved similar morphology and ecological niches to toucans[32], but there is no

evidence that they use their feet to manipulate objects. Another example is that foot use evolved in relation to eating invertebrates in Corvides and Sylvioidea (Fig. 2a, Supplementary Table 3), but many other clades of passerines and Telluraves have diets based on invertebrates yet do not use their feet to hold or manipulate prey. In fact, several bird groups have evolved alternatives to using their feet to handle and feed on invertebrate and vertebrate prey. For example, trogons, coraciiform birds (e.g., kingfishers, motmots, bee-eaters), and roadrunners (*Geococcyx*, Cuculiformes) that feed on large insects and small vertebrates grasp their prey with the beak and beat and shake it repeatedly against a surface to kill and clean the prey before eating it whole[33]. This same behavior can also be observed in many suboscines that feed

on large insects, such as antbirds (Thamnophilidae)[34] and fly-catchers (Tyrannidae)[35]. The evolution of skilled foot use in birds is therefore likely to be a product of diet (insects, small verte-brates), foraging or capture method (e.g., gleaning vs hawking), and morphology, but the relative contributions of each remain uncertain.

A key component of skilled manipulation in mammals is the ability to grasp and bring objects to the mouth[1,3]. Although the transition to arboreality has resulted in the repeated evolution of foot use among birds, the ability to grasp, and particularly to bring objects to the beak, has evolved only a few times. This feature is restricted to owls, falcons (Fig. 3), mousebirds, parrots (Fig. 4) and three families within the suborder Corvides (Fig. 5b). As discussed above, the evolution of foot use is related to several different diets and not necessarily in a consistent fashion. How-ever, there is some evidence to suggest that the evolution of pedal grasping and bringing objects to the beak has its origin with a predatory diet. While our ancestral state reconstruction recovers granivory and frugivory as the likely ancestral states of extant parrots and mousebirds respectively (Supplementary Table 3), there is evidence that extinct, stem branches of both of these clades had a more predatory diet. Most stem group Coliiformes, as well as an extinct sister clade (Sandcoleidae) had morpho-logical traits, such as shortened proximal phalanges in the second and fourth toe[36], that suggest they were adapted to capture and manipulate large objects, including prey. Also, stem Coliiformes had proportionally longer beaks than extant Colliformes, as well as other beak adaptations, that suggest a less strictly frugivorous diet[36,37]. Similarly, fossil evidence suggest that two of the stem pan-Psittaciformes clades had many raptor-like adaptations in the beak and foot[38–40]. A raptorial diet for stem parrots is also supported by enhanced fat digestion and absorption in parrots, which is shared with the three raptorial orders[41]. Thus, while the ancestor of extant Psittaciformes and Coliiformes was likely a frugivore and/or granivore (Supplementary Table 3), it is possible that the earlier ancestors of both clades had a raptorial ecology that included the ability to grasp and bring objects to the beak, and that only later was this ability exapted for the manipulation of fruit and seeds. Nevertheless, it is possible that the ability to grasp and bring objects to the beak has evolved independently, and for different reasons in different groups of birds. In the case of parrots for example, it is possible that the combination of a zygodactyl foot, which allows for a firm grip[38], and a diet based on extractive foraging of seeds and fruits[13] is what drove the evolution of this behavior.

The evolution of skilled manipulation in mammals is asso-ciated with changes to sensory and motor circuits, as well as adaptations of the skeleton and integument of the manus[42]. In the somatosensory system, mammals that use their forelimbs for haptic searching and complex object manipulation evolve glab-rous skin and an increase in the number, acuity and sensitivity of touch receptors[43]. Touch receptors can be found in avian feet[44] and share some similarities with receptors in the mammalian manus. For example, the plantar skin of owls has specialized tubercles, each of which contains a Herbst corpuscle (the avian equivalent of the mammalian Pacinian corpuscle), and the claw has a dual and detailed topographic representation in the anterior Wulst, which is equivalent to the somatosensory cortex in birds[45]. Based on this, one would predict the same to be true of falcons, hawks, and parrots, but currently this is unknown. In the motor system, differences in forelimb manipulation skills among mammals are also correlated with anatomical changes. Whereas most mammals capable of manipulating objects with the hands have a corticospinal tract (direct projections from the cortex to the spinal cord), this is particularly developed in primates, where both ipsilateral and contralateral projections are present, and

cortical projections make direct contact with motoneurons in the spinal cord[2]. An equivalent "corticospinal" projection does not appear to be well developed in birds[46], although many aspects of motor control in birds remains understudied. The closest to a corticospinal tract in birds has been found in the zebra finch (Taeniopygia castanotis), where the anterior Wulst projects to the spinal cord, but these fibers reach only to the level of C7, and are sparse[47]. Corticospinal projections to the cervical spinal cord have also been suggested in owls[48], but in parrots, despite the high manipulative skills with the feet, there does not appear to be any direct projections from the Wulst or other parts of the pal-lium to spinal cord regions associated with hindlimb movements[46]. Whether other birds that use their feet to manip-ulate objects have direct projections to the spinal cord is unknown. In fact, almost nothing is known about the control of voluntary movement in birds; it even remains unclear if birds possess a region that functions similarly to mammalian primary motor cortex[48].

Despite the gaps in our understanding of avian somatosensory and motor systems, it is clear that skilled manipulation has evolved in parallel in mammals and birds. In addition, arboreality and diet appear to have played a role in the evolution of skilled manipulation in birds, as has also been suggested for mammals. Owing to these parallels, further research into skilled hindlimb use in birds will provide new insights into the neural basis of skilled limb use more generally and may also aid in the inter-pretation of the behavior of extinct species. In birds, it is well established that both an arboreal niche and a raptorial diet are correlated with the morphology of the pedal phalanges[4,49], which has been used extensively to infer the raptorial lifestyle (and foot use) of fossil birds and non-avian dinosaurs[37]. Nonetheless, it is unclear whether other diets or how foot use can be predicted from pedal morphology. The data presented here on the dis-tribution of foot use along the avian phylogeny and the differ-ences between clades in manipulative skills may set the stage for more accurate comparisons of pedal morphology with behavior and diet in birds thereby allowing for better prediction of the ecology and behavior of extinct species.

## Materials and methods

**Database construction.** To study manipulation of objects with the foot in birds, we combined exhaustive searches of media in clades known to use their feet with extensive literature searches to determine which clades use their feet to manipulate objects. To determine which avian families exhibit skilled foot use, we first per-formed a literature search for reports of foot use. This included a systematic search of *Birds of the World*[50] for any report of foot use. Additionally, we performed full text searches of ornithological journals in the Biodiversity Heritage Library (BHL) for the phrases "a foot," "under a foot," "its foot," "its feet," "held under a foot," and others. Along with previous reviews of the literature[8,51], we were able to collect references on foot use in 259 species of birds belonging to 85 families (Supple-mentary Table 7). However, some of these reports are anecdotal, one-time occurrences, or even mistaken. For example, Clark[8] cited Skutch[52] as evidence of foot use in trogons, but the cited reference is about a toucan species that uses its feet to manipulate objects. Other reports, such as that of the tooth-billed pigeon (Didunculus strigirostris), seem to be a repetition of a single report without any supporting evidence[53]. To remedy this, for each of the species reported in the literature as using their feet to manipulate objects, we performed an in-depth search for media (described below). Because some of these species have very few pictures available (like the endangered tooth-billed pigeon, which has 0 in the Macaulay library and only a few outside of it), if no pictures of the species reported as using their feet were found, we then searched for foot use in additional species in the same family. To maximize the probability of finding foot use, we searched the top 5% of species with the most pictures in the Macaulay Library within that family. With this method we were able to confirm foot use in 59 of the 85 families where foot use has been reported. In total there were 26 families where foot use had been reported in the literature, but for which we could find no media showing foot use. These are families where foot use is either very rare or misreported. The former applies to families like herons (Ardeidae) and flycatchers (Tyrannidae), where a few species have been reported to use their feet, but no photos or videos of foot use were found. These families were not considered as having the ability to use their feet to manipulate object in our analysis. With this method, it is possible that

we missed some species that use their feet to grasp or manipulate objects, but it is likely that these are few and do not belong to any clade where foot use is widespread.

To quantify and compare pedal manipulation skills across species, we searched for videos or pictures of foot use associated with food or other types of object manipulation, an approach similar to previous studies[54]. Based on previous literature[8,51] we first systematically searched all species of those orders or families in which foot use is widely reported. These included all diurnal and nocturnal raptors (Strigiformes, Falconiformes, and Accipitriformes), seriemas (Cariamiformes), parrots (Psittaciformes), mousebirds (Coliiformes), and swamphens (genus *Porphyrio* and Gruiformes). We also systematically searched three closely related families in the order Piciformes: Ramphastidae (toucans), Capitonidae (New-World barbets) and Semnornithidae (toucan-barbets). Within the songbirds (Passeriformes), we systematically searched for six families in which foot use is widely reported: Corvidae (crows, jays, and allies), Paridae (tits, chickadees, and allies), Druridae (drongos), Icteridae (blackbirds, caciques, and grackles), Estrilidae (finches), and Laniidae (shrikes and allies). For the list of species we use the taxonomy of birdtree.org[55].

The sites used to search for photographs or videos were Macaulay Library (https://www.macaulaylibrary.org), Wikiaves (https://www.wikiaves.com.br), Google Images, Flickr, Alamy, Youtube, Twitter, and Instagram. We always searched pictures first in the Macaulay Library because it is a curated source for species identification and contains all bird species in the world. Up to 2000 media files (pictures and videos) were examined for each species. Only ~10% of the species in the Macaulay Library have >2000 media files (as of 2021, Supplementary Fig. 4e), which means that for 90% of the species examined, we looked at all media available in this data repository. The second main site was Wikiaves. It is also a curated source for species identification, but contains only Brazilian bird species. Then the species were searched on the following sites in this order: Google Images, Flickr, Alamy, YouTube, Twitter, and Instagram. The keywords used to search on these sites were: "Latin name", "English name", "Latin name + feeding", "English name + feeding", "Latin name + eating", and "English name + eating". Species with few photos (<500) in the first two sites were systematically searched using all keywords.

**Detection thresholds**. An important issue was to distinguish between species that do not use their feet for manipulation from those that do, but where only a limited number of photos and videos are available and therefore with a lower likelihood to detect the behavior. To address this problem, we estimated detection thresholds based on media availability. This requires knowing the number of files available for each species, which are published annually by the Macaulay Library (https://www.macaulaylibrary.org/resources/media-target-species/). For all our calculations, we used the March 2021 update because it is the closest to the dates when our searches occurred. We used the sum of all pictures and videos available for each species. First, we tested whether our skill index (see below) correlated with the number of media items. Although one could expect that more pictures would be associated with a higher number of different behaviors and therefore a higher skill index, we found no significant correlation (PGLS, $F_{1,1018} = 2.877$, $p$-value = 0.26, Supplementary Table 2, Supplementary Fig. 4c). We then calculated a "detection probability" for each species at the clade level (i.e., family or order), which represents the detection threshold of the number of media at which there was a 75% chance of detecting foot use (Supplementary Fig. 4a, b). For this purpose, we fitted a logistic regression for each clade with the number of media as the predictor using the ggplot[56] package in R[57].

We then used the lower 95% confidence interval of that curve to determine the number of media at which the 75% probability of detection threshold was crossed. In the case of parrots, because some species do not use their feet at all, we removed all species that had >500 media, but no foot use, to calculate the threshold. Supplementary Table 4 shows the calculated threshold for each systematically searched clade. The 75% threshold varies greatly among clades and was as low as 77 photos/videos in parrots and as high as ~2700 for some of the songbird families. We then used this threshold (rounded up conservatively, Supplementary Table 4) to determine in which species there were not enough media entries to determine if foot use was present (media below the threshold) and in which foot use was not present (media above the threshold). In the case of the Laniidae and Dicuridae, the number of species is too small to fit a logistic regression so we used a threshold calculated for the Corvoidea.

**Prehensile behavior taxonomy and scoring**. To record and quantify manipulative behavior with the feet, we followed previous research in mammals[15] and calculated a pedal dexterity index. Table 1 and Fig. 1a show the 10 behaviors that were recorded. In each picture or video, we recorded the presence of any of these behaviors and assigned a score of one. Dataset1 shows the scores for each individual picture or video. A dexterity index for each species (Dataset 2) was then computed as the sum of all the behaviors present for each species, so the dexterity index for each species was a number between 0 and 10. Figure 1a shows three examples of this scoring. This scoring system allow us to compare the general skill level across species as well as the presence or absence of specific behaviors, such as grasping or the ability to lift objects to the beak.

**Phylogenies and ancestral state reconstruction**. For the family-level analysis, we used a family-level tree generated by Toda et al.[58]. Briefly, in this tree the

backbone is based on a phylogenomic supertree[59]. Relationships for passerines are based on Oliveros et al.[23]. Family names correspond to Clements[60]. Families missing in the Kimball et al.[59], backbone (Chionidae, Pluvianellidae, Pluvianidae, Ibidorhynchidae and Stercorariidae) were added using a midpoint rooting method implemented in addTaxa[61], based on their phylogenetic position[62]. Other families with more uncertain positions, such as Semnornithidae and Teretistridae, were also included using the midpoint rooting method implemented in addTaxa[63,64]. To build genus-level trees for owls (Strigiformes), hawks, eagles, vultures (Accipitriformes), and falcons (Falconiformes) we extracted 1,000 fully resolved trees from birdtree.org[55], and built a maximum clade credibility (MCC) tree using *phangorn*[65]. The same procedure was used for the passerine families shown in Fig. 5 and Supplementary Fig. 3. We then pruned each tree until only one species per genus remained. In the case of parrots (Psittaciformes) the phylogeny used was a recently published MCC consensus phylogeny[66]. Ancestral state reconstruction of foot use was performed on our family-level phylogeny, as well as on the maximum clade credibility tree at the species level for parrots (Supplementary Tables 1, 5), using the R package *corrHMM*[67], which implements a maximum-likelihood method that allows multi-state characters and polymorphic taxa. We then compared the fit of three different models: (1) an all-rates different matrix (ARD) model, in which all possible transitions between states receive distinct values; (2) a one-parameter equal rates (ER) model, in which a single rate is estimated for all possible transitions; (3) a symmetric (SYM) model in which forwards and reverse transitions between states are constrained to be equal. In the family-level reconstruction we also tested a fourth model, the precursor model of Marazzi et al.[68]. In this precursor model (PREC), the observed state (foot use) could be exhibited only by a lineage that had transitioned from no foot use to the precursor state first. Thus, transitions from the no foot use directly to foot use were prohibited. In the family-level ancestral state reconstruction, families in which not enough data was available to assess the presence or absence of foot use were entered as unknown, which assigns an equal probability to both characters. Families entered as not enough data were those in which the total number of media entries for the whole family was <1500. This threshold was based on the upper range of detection thresholds calculated for individual orders and families (see above, Supplementary Fig. 4a, b, Supplementary Table 4). Because we found that at the family-level, the number of pictures per family in the Macaulay Library is strongly correlated with the research effort for each family (Supplementary Fig. 4d), this threshold not only reflects the probability of detect foot use behavior through pictures/video, but also in literature reports. Research effort for 8648 species of birds was obtained from Ducatez et al.[69], which corresponds to 206 of the 249 families of birds. Research effort and total number of media were added for each family.

Diet information for all species was obtained from Pigot et al.[32], which used an updated version of the EltonTraits dataset[70,71]. While updated, the ecology and diet of many birds species remains poorly known and therefore some of this data is inevitably inaccurate and may change in the future. In the Pigot et al.[32], dataset, diet (trophic niche) is a categorical character in which a species is considered to belong to particular niche if >60% of the diet is of one type. Omnivores are species where no diet is >60%. These categories are shown in Fig. 2 and Supplementary Fig. 2 for parrots. We also used these categories to reconstruct the ancestral diets of the main groups in which foot use was expressed. For simplicity we used a MCC tree from[72] but that also correspond to the Hackett backbone from birdstrees.org. Ancestral state reconstruction was also performed with corHMM, and, as described above, the best fitting model based on AIC values was used to extract ancestral diets (Supplementary Table 2). Because omnivory can be prevalent in some clades, we also used the original percentage base data from the EltonTraits[70] to reconstruct ancestral percentage of a given diet. For New-World barbets and toucans, we reconstructed the percentage of fruit and vertebrates (Supplementary Fig. 1c, d), while in parrots we reconstructed the percentage of seeds, fruit and nectar (Supplementary Fig. 2b-d). In this case, we treated diet as a continuous character and used the *contmap* function in the package phytools[73] to fit this character and ancestral states to the corresponding phylogenies. Phylogenetic Generalized Least Squares (PGLS) were performed using the R packages *ape*[74] and *nlme*[75].

**Images of birds using their feet**. All images of birds using their feet were from the Macaulay Library at Cornell University. Catalog numbers and photographer credits are listed in Supplementary Table 6.

**Reporting summary**. Further information on research design is available in the Nature Portfolio Reporting Summary linked to this article.

## Data availability
All data are included in the electronic supplementary materials and Supplementary Data 1, 2.

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

# ARTICLE

60. Clements, J. F. et al. *Downloadable Checklist | Clements Checklist*. https://www.birds.cornell.edu/clementschecklist/download (2022).

61. Mast, A. R. et al. Paraphyly changes understanding of timing and tempo of diversification in subtribe Hakeinae (Proteaceae), a giant Australian plant radiation. *Am. J. Bot.* **102**, 1634–1646 (2015).

62. Fain, M. G., Krajewski, C. & Houde, P. Phylogeny of "core Gruiformes" (Aves: Grues) and resolution of the Limpkin–Sungrebe problem. *Mol. Phylogenet. Evol.* **43**, 515–529 (2007).

63. Moyle, R. G. Phylogenetics of barbets (Aves: Piciformes) based on nuclear and mitochondrial DNA sequence data. *Mol. Phylogenet. Evol.* **30**, 187–200 (2004).

64. Barker, F. K., Burns, K. J., Klicka, J., Lanyon, S. M. & Lovette, I. J. Going to extremes: contrasting rates of Diversification in a recent radiation of New World passerine birds. *Syst. Biol.* **62**, 298–320 (2013).

65. Schliep, K. P. phangorn: phylogenetic analysis in R. *Bioinformatics* **27**, 592–593 (2011).

66. Smith, B. T. et al. Phylogenomic analysis of the parrots of the world distinguishes artifactual from biological sources of gene tree discordance. *Syst. Biol.* https://doi.org/10.1093/sysbio/syac055 (2022).

67. Boyko, J. D., Beaulieu, J. M., Oliver, J. & Boyko, J. *corHMM 2.1: Generalized Hidden Markov Models. R Package Version 2.8.* https://rdrr.io/cran/corHMM/ (2022).

68. Marazzi, B. et al. Locating evolutionary precursors on a phylogenetic tree. *Evolution* **66**, 3918–3930 (2012).

69. Ducatez, S., Sol, D., Sayol, F. & Lefebvre, L. Behavioural plasticity is associated with reduced extinction risk in birds. *Nat. Ecol. Evol.* **4**, 788–793 (2020).

70. Wilman, H. et al. EltonTraits 1.0: Species-level foraging attributes of the world's birds and mammals. *Ecology* **95**, 2027–2027 (2014).

71. Tobias, J. A. & Pigot, A. L. Integrating behaviour and ecology into global biodiversity conservation strategies. *Philos. Trans. R. Soc. B: Biol. Sci.* **374**, 20190012 (2019).

72. Tobias, J. A. et al. AVONET: morphological, ecological and geographical data for all birds. *Ecol. Lett.* **25**, 581–597 (2022).

73. Revell, L. J. phytools: an R package for phylogenetic comparative biology (and other things). *Methods Ecol. Evol.* **3**, 217–223 (2012).

74. Paradis, E. et al. Package 'ape'. *Anal. Phylogenet. Evol.* **2**, 47 (2019).

75. Pinheiro, J. et al. Package 'nlme'. *Linear and Nonlinear Mixed Effects Models, Version 3*, https://cran.r-project.org/web/packages/nlme/index.html (2017).

## Acknowledgements

We would like to thank Dr. Maude Baldwin for editorial comments. We would also like to thanks Prof. Taichi Kato for finding many additional media of raptors using their feet. Funding for this work was obtained from The Natural Sciences and Engineering Research Council of Canada (NSERC) by D.R.W. C.A.-P. received a scholarship from the National Council for Scientific and Technological Development (CNPq), Brazil.

## Author contributions

C.G.-I., C.A.-P. and J.B. designed the research. C.G.-I. and C.A.-P. collected and analyzed behavioral data, and prepared figures. J.B. and D.R.W. acquired funding. C.G.I., C.A.-P. and A.N.I. wrote the manuscript with input from all authors.

## Competing interests

The authors declare no competing interests.
