## [Peer Review File · Communications Biology]

Reviewers' comments:

Reviewer #1 (Remarks to the Author):

This manuscript presents a novel, comprehensive phylogenetically-informed analysis of a citizen-science repository of images and videos involving foot usage in an impressive diversity of avian taxa. The use of feet in birds has been studied very little (certainly in comparison to flight), and this work provides a refreshing outlook on other ways in which birds are amazingly functionally diverse. The authors sought to quantify "skilled foot use" for as many avian taxa as possible, and to test the hypothesis that arboreality was a precursor for "skilled foot use" in birds. They also examined the importance of diet as a putative driver of skilled foot use. The authors found that arboreality is in fact a precursor for skilled foot use in birds (evolving once at the base of Telluraves), and that skilled foot use has little relation to diet (except for within the Corvides, for instance). They found very interesting patterns within certain clades, such as multiple independent origins in some clades, and multiple losses of skilled foot use in others, such as the parrots. One thing that I found surprising was how restricted the evolution of grasping and bringing items to the beak actually is, having evolved only a few times across a few different clades, which points to a predatory diet origin.

I felt that the manuscript was very well organized and written very clearly and concisely. I found no discernable flaws in the design/execution of the methods or presentation of the results, and the statistical analyses all seemed very sound to me. The figures and graphics are very nicely composed and all very informative. Below I list a few points where I struggled, and I make a few suggestions that the authors should feel free to do with as they please.

General Comments

1. I'm not sure I would call this a "citizen science approach". The use of a public database for research and publication of the results is certainly creative, resourceful, and laudable, but it's not the "citizens" who actually conducted the "science" here. When I think of "citizen science", I think of recruiting the public to help gather data for analysis; in this case all of the data (foot use scores) were gathered by the authors. I suppose, however, one could argue that uploading photos to the internet is "data gathering" in a way.

2. I really appreciated the steps that the authors took to verify foot usage for each species from the images available. However, I was unable to fully follow the calculation of detection probability (e.g., lines 391-394). I trust that the authors used this tactic for a reason, but it was not intuitive to me. For example, how many "rolling bins" were analyzed for each clade? I might have used a logistic regression to calculate the probabilities directly from the presence/absence (1,0) of foot use, and the number of images available for each species. Along these lines, regarding the null correlation between skill index and number of pictures (lines 386-388); if these include ALL photos – including those for which there was no opportunity to assess foot use if present – then this would negatively bias the correlation. It seems to me that a stronger argument would be made if only those photos depicting some kind of feeding or nesting behavior were included– where the possibility of detecting foot use, if it exists in a species, is tenable.

I also trust that the authors employed their "75%" detection criterion for a reason, although some might wonder if/how the results might differ with a stiffer 90% criterion?

3. I wondered about the overall effect of body size in skilled foot use; the analyses did not explicitly appear to incorporate body size as far as I could tell, although it was addressed briefly in the context of raptors.

4. The ancestral state reconstruction analyses were very well explained, but some of the other PGLS implementations were not; in some cases it was not clear to me what the dependent and/or independent variables were, fixed or random effects, model error distributions, parameter estimates, etc. I know a lot more goes into, and comes out of, running these models, and the authors might consider showcasing this in the supplemental materials.

Specific Comments

Line 70: Regarding the references listed at the end of the sentence in this line, it seems to me that "9" should go with the previous statement.

Lines 170-171: Consider finishing the thought here; "The smaller mass of falcon and owls is reflected in a much higher percentage of species within these groups that feed on invertebrates"... which explains the relatively higher skill indices of these groups?

Lines 93-94: This approach of scoring images for foot usage from the internet is similar to that used by:

Sustaita D, Gloumakov Y, Tsang LR, Dollar AM. 2019. Behavioral correlates of semi-zygodactyly in Ospreys (*Pandion haliaetus*) based on analysis of internet images. PeerJ 7:e6243 DOI 10.7717/peerj.6243. <https://peerj.com/articles/6243/>

Line 486: "where" should be 'were'

Supplementary figure 4. a, and b caption: I think "interest" should be 'intersect'

Thank you for the opportunity to review this wonderful and insightful manuscript!

-Diego

Reviewer #2 (Remarks to the Author):

I very much enjoyed reading this manuscript and believe that it is well suited for Communications Biology.

While the phylogeny and mechanism of dexterity have been well researched in primates, there are only few studies on foot use in birds and none on the phylogeny of the latter (to my knowledge). This is unfortunate as the emergence of similar behaviors in distantly related species such as birds and primates can be highly telling on the driving forces underlying the convergent evolution of dexterity. Gutierrez-Ibanez et al., have conducted an impressive data search, through online repositories and applied previous knowledge to selectively analyzed nearly 4000 media files. They identified, categorized, and rated the complexity of foot use (if present) across 64 families of birds. They additionally targeted associations of the animal's niches and foraging styles.

I believe that this study has the potential to become a valuable reference for future studies on foot use. In addition to this it may also be useful to comparative psychology which is increasingly placing birds in direct comparison to primates on physical problem-solving behavior.

A media search of this dimension is bound to have some limitations such as comparatively few or no pictures being available for some groups, misleading pictures etcetera. Nevertheless, I feel that the authors are acknowledging and/or controlling for the latter to a sufficient extent for the outcomes still bear substantial value. I do have a some comments and suggestion that I list below in the order in which they appear.

Title: I would consider removing or replacing the word 'skilled' in the title. To me a skill is applying

knowledge or using practice to do something well. Since the authors are targeting pre-existing behaviors in this paper I suggest they replace it with something like 'dexterous' or remove it.

Lines 56-59 It would be interesting to elaborate a bit more on role of manipulation in primate brain evolution.

Lines 66 I think that wings have more functions than just flight (such as regulating body temperature, displays and in some cases aquatic movement).

Line 73 'Phylogeny' may fit better than 'evolution' at this spot.

Line 75 It may be interesting to elaborate more on the species sample used in this study.

Line 133 You might add here that some parrots do also use their feet during problem solving.

Lines 161-166 Concerning the loss of grasping in the Old-World vultures: I have little expertise on this but perhaps the authors can clarify: from what I understand New World vultures might actually be assorted to

Accipitriformes ? This seems also true for the extinct Tetrathornithidae which were likely to be scavengers as well. The question is: If the order includes one mixed, two predatory, two scavenger families could we still safely assume that the ancestral state is predatory?

Lines 186-189 The way parrots grasp object is not incorrectly described but I found it a bit misleading in the text and in the Figure. It is true that the foot is being turned inward in those birds relative to their ankle bones. I feel when we think about turning a limb inward while grasping we (being primates) tend to intuitively think about the orientation of the palm. As in most parrot the palm is oriented outwards (but inwards in Amazon and Eclectus) while grasping I found this confusing.

Line 233 I would recommend replacing the word 'confirm' with 'support'.

Lines 249-252 Shrikes and drongos also feed on small vertebrates and eat them up in the trees. Could this combination foster lifting behavior? Note perhaps also that many insectivorous birds swallow their prey in whole and do not have to process it first.

Lines 269-282 It would be nice to provide some alternative explanations. Notably, parrots also have other traits (than possibly having meat-eating ancestors) that may have fostered foot use. Their zygodactyl feet are probably an adaption to clambering and climbing in the canopy, but they could arguably be better for grasping than anisodactyl feet. Parrots dehusk complex and often large seeds with their beak. This can be a tedious process in which the seed has to be turned in between manipulation bouts. With larger seeds this is not always possible to keep them in the beak while turning them. Note also that many parrots do opportunistically feed on insects and small vertebrates.

Line 330 Phrases like or exactly those phrases?

Lines 452-454 You might mention that still little is known about the details of the wildlife ecology of many avian species and that some niches may thus not be up to date.

Table 1: This needs some clarifications. What is the difference between 'use of toes' and 'grasping'? what is the difference between 'use of one leg and free grasping (without flight)'?

Reviewer #1 (Remarks to the Author):

This manuscript presents a novel, comprehensive phylogenetically-informed analysis of a citizen-science repository of images and videos involving foot usage in an impressive diversity of avian taxa. The use of feet in birds has been studied very little (certainly in comparison to flight), and this work provides a refreshing outlook on other ways in which birds are amazingly functionally diverse. The authors sought to quantify “skilled foot use” for as many avian taxa as possible, and to test the hypothesis that arboreality was a precursor for “skilled foot use” in birds. They also examined the importance of diet as a putative driver of skilled foot use. The authors found that arboreality is in fact a precursor for skilled foot use in birds (evolving once at the base of Telluraves), and that skilled foot use has little relation to diet (except for within the Corvides, for instance). They found very interesting patterns within certain clades, such as multiple independent origins in some clades, and multiple losses of skilled foot use in others, such as the parrots. One thing that I found surprising was how restricted the evolution of grasping and bringing items to the beak actually is, having evolved only a few times across a few different clades, which points to a predatory diet origin.

I felt that the manuscript was very well organized and written very clearly and concisely. I found no discernable flaws in the design/execution of the methods or presentation of the results, and the statistical analyses all seemed very sound to me. The figures and graphics are very nicely composed and all very informative. Below I list a few points where I struggled, and I make a few suggestions that the authors should feel free to do with as they please.

General Comments

We like to thank both reviewers for their kind comments, thorough review and detailed comments. Next we provided detailed response to each comment.

1. I'm not sure I would call this a “citizen science approach”. The use of a public database for research and publication of the results is certainly creative, resourceful, and laudable, but it's not the “citizens” who actually conducted the “science” here. When I think of “citizen science”, I think of recruiting the public to help gather data for analysis; in this case all of the data (foot use scores) were gathered by the authors. I suppose, however, one could argue that uploading photos to the internet is “data gathering” in a way.

We agree with the reviewer that the most traditional definition of “citizen science” implies the direct involvement of “citizens” in collecting and even analyzing data. On the other hand, we are not the first researcher to call pictures depositories like the Macaulay library and wikiaves a “citizen science” approach. In the past few years several studies have use these depositories to study both behavior and morphology, and have called it “citizen science” (e.g. (Berryman and Kirwan 2021; Tubelis and Sazima 2021; Pyle 2022; Lopes and Schunck 2022)). In this sense at the very least, we are not the first to call data collected from these depositories a citizen science approach.

We would argue that in these cases, by uploading pictures the “citizens” are collecting a myriad of data, just not anything specific. But given the exponential growth of bird pictures in these depositories we think that it is only fair to call them “citizen scientist”. In conclusion, we would like to keep the title as it is.

2. I really appreciated the steps that the authors took to verify foot usage for each species from the images available. However, I was unable to fully follow the calculation of detection probability (e.g., lines 391-394). I trust that the authors used this tactic for a reason, but it was not intuitive to me. For example, how many “rolling bins” were analyzed for each clade? The rolling bin was 10 species, so the number of bins would be equal to the number of species clade minus 20. (you can't calculate the probability for the first and last 10 species. I might have used a logistic regression to calculate the probabilities directly from the presence/absence (1,0) of foot use, and the number of images available for each species.

We thank the reviewer for this comment. We have now used, as suggested, a logistic regression to determine the 75% detection threshold. To reflect this, we have changed the corresponding section in the methods (starting at line 388) as well as figure S4a-b and corresponding figure caption (supplementary materials) which now shows logistic regression curves for each clade. Finally, this also changed Table S4. Notice that the calculated thresholds were in most cases very similar (e.g. Accipitiformes changed from 108 to 123, Parrots from 66 to 77). In some cases, like Estrildidae, the threshold increased significantly, from 1700 to 2400, but because our analysis was for most clades at the genus level this did not change the inclusion of any genus nor in having to change any analysis or figures.

Along these lines, regarding the null correlation between skill index and number of pictures (lines 386-388); if these include ALL photos – including those for which there was no opportunity to assess foot use if present – then this would negatively bias the correlation. It seems to me that a stronger argument would be made if only those photos depicting some kind of feeding or nesting behavior were included– where the possibility of detecting foot use, if it exists in a species, is tenable.

We don't think this would create a strong bias because, as we point out in the methods (Fig S4e), only 10 % of all species have more than 2000 photographs (at least at the time the analysis was done). This means that for 90 % of species we did look at all the pictures available in the Macaulay library. Also, we don't think that there is *a priori* reason to think that the number of pictures feeding or nesting are over or underrepresented in a group, and therefore they should be proportional to the total media, and thus the resulting correlation (or lack thereof) should be the same. Finally, we would like to point out that counting the number of pictures that depict feeding would be a monumental task as in the Macaulay library only a small amount of them are actually labeled and can be filtered automatically. Counting them manually would take a considerable effort and doubtfully a different outcome.

I also trust that the authors employed their “75%” detection criterion for a reason, although some might wonder if/how the results might differ with a stiffer 90% criterion?

A change in threshold to 90 % would have changed the results or analysis very little. For example, in the case of parrots a change from 75% to 90 % would have resulted in a threshold change from 77 to 269 but this would only result in the exclusion of two additional species as undetermined and not included as no foot use, and this two species belong to genera where other species with more media also showed no foot use. In the case of owls, the threshold changes from 1000 to 3000 but only one species with more than 1000 media showed not foot use, and our analysis was at the genus level, so again this would result in no changes to the overall results or analysis.

3. I wondered about the overall effect of body size in skilled foot use; the analyses did not explicitly appear to incorporate body size as far as I could tell, although it was addressed briefly in the context of raptors.

This is an interesting question but we did not include body mass in our analysis as we think is beyond the scope of our manuscript which already cover a large amount of data but also because our data indirectly suggest that there is no clear pattern. In the case of raptors for example, as alluded in our manuscript, smaller species seems to show more skill and very large species (vultures) show a loss of foot use skill. In contrast, in the case of parrots, and as we alluded briefly in the discussion (line 182-183), is mostly small clades that have lost foot use. Therefore, when it comes to skilled foot use, there seems to be a complex interaction between body size, diet and phylogeny. Perhaps future studies that look at specific clades in more detail can further dissect the effect of body size on foot use.

4. The ancestral state reconstruction analyses were very well explained, but some of the other PGLS implementations were not; in some cases it was not clear to me what the dependent and/or independent variables were, fixed or random effects, model error distributions, parameter estimates, etc. I know a lot more goes into, and comes out of, running these models, and the authors might consider showcasing this in the supplemental materials.

We thank the reviewer for this suggestion. To make the models clearer, we have added a table to the supplementary materials (Table S2 now) with more details of each of PGLS. This includes the values and confident intervals for the lambda estimates. In all cases this are simple models with only one dependent variable. The addition of this table has change the number of all supplementary tables, which have has been changed in the main text and supplementary materials.

Specific Comments

Line 70: Regarding the references listed at the end of the sentence in this line, it seems to me that "9" should go with the previous statement.

This is correct. We have changed the position of the reference to the previous sentence.

Lines 170-171: Consider finishing the thought here; “The smaller mass of falcon and owls is reflected in a much higher percentage of species within these groups that feed on invertebrates”... which explains the relatively higher skill indices of these groups?

We have changed this and now reads:

“The smaller mass of falcon and owls is reflected in a much higher percentage of species within these groups that feed on invertebrates (Fig. 3f), which likely explains the higher skill indices of these groups compared to accipitriforms

Lines 93-94: This approach of scoring images for foot usage from the internet is similar to that used by:

Sustaita D, Gloumakov Y, Tsang LR, Dollar AM. 2019. Behavioral correlates of semi-zygodactyly in Ospreys (*Pandion haliaetus*) based on analysis of internet images. PeerJ 7:e6243 DOI 10.7717/peerj.6243. <https://peerj.com/articles/6243/>

We have added this reference to our methods sections, line 353.

Line 486: “where” should be ‘were’

We have made this change in line 468.

Supplementary figure 4. a, and b caption: I think “interest” should be ‘intersect’

We have change it to intersect.

Thank you for the opportunity to review this wonderful and insightful manuscript!
-Diego

Reviewer #2 (Remarks to the Author):

I very much enjoyed reading this manuscript and believe that it will ultimately be well suited for Communications Biology. While the phylogeny and mechanism of dexterity have been well researched in primates, there are only few studies on foot use in birds and none on the phylogeny of the latter (to my knowledge).

This is unfortunate as the emergence of similar behaviors in distantly related species such as birds and primates can be highly telling on the driving forces underlying the convergent evolution of dexterity. Gutierrez-Ibanez et al., have conducted an impressive data search, through online repositories and applied previous knowledge to selectively analyze nearly 4000 media files. They identified, categorized, and rated the complexity of foot use (if present) across 64 families of birds. They additionally targeted associations of the animal's niches and foraging styles.

I believe that this study has the potential to become a valuable reference for future studies on foot use. In addition to this it may also be useful to comparative psychology which is increasingly placing birds in direct comparison to primates on physical problem-solving behavior.

A media search of this dimension is bound to have some limitations such as comparatively few or no pictures being available for some groups, misleading pictures etcetera. Nevertheless, I feel that the authors are acknowledging and/or controlling for the latter to a sufficient extent for the outcomes still bear substantial value. I do have a few comments and suggestions that I list below in the order in which they appear.

1. Title: I would consider removing or replacing the word 'skilled' in the title. To me a skill is applying knowledge or using practice to do something well. Since the authors are targeting pre-existing behaviors in this paper I suggest they replace it with something like 'dexterous' or remove it.

We appreciate the suggestion by the reviewer but "skilled" has been used extensively to describe the reaching and grasping movements of

2. Lines 56-59 It would be interesting to elaborate a bit more on role of manipulation in primate brain evolution.

We have added a sentence to expand on this matter, it now reads:

"Because grasping and manipulating objects are characteristic of humans and nonhuman primates, the neural basis of these behaviors, and their association with primate evolution, including the brain, have received extensive attention¹⁻⁴. **In the case of primates brain evolution, the development of skilled manipulation has been related to the evolution of specialized visual and motor circuits^{2,3,5}.**"

3. Lines 66 I think that wings have more functions than just flight (such as regulating body temperature, displays and in some cases aquatic movement).

The reviewer is correct that wings have some other functions, we have changed this to better reflect this fact. It now reads:

“In birds, forelimbs have gained the **almost** exclusive function of flight, with grasping transferred predominantly to the beak^{5,6}

4, Line 73 ‘Phylogeny’ may fit better than ‘evolution’ at this spot.

We respectfully disagree with the reviewer here but to make the phrase clearer we have change it and now reads:

“A broader approach is required to establish if the development of manipulative foot use aligns with the evolution of arboreality in birds.”

5. 75 It may be interesting to elaborate more on the species sample used in this study.

We have added the number of species in the studied referenced in this line. It now reads:

... arboreality predates manipulative foot use in birds, although this was based on only a limited species sample (**around 150 species**)

6. Line 133 You might add here that some parrots do also use their feet during problem solving.

We have changed this to reflex the more uses of manipulation with the foot in parrots it now reads:

“This is driven by the capacity of most parrots to grasp and bring object to the beak while also rotating their foot to manipulate objects, **which includes not only food items but also tools and others non-food objects^{24,25}**.”

7. Lines 161-166 Concerning the loss of grasping in the Old-World vultures: I have little expertise on this but perhaps the authors can clarify: from what I understand New World vultures might actually be assorted to Accipitriformes ? This seems also true for the extinct Tetrathornithidae which were likely to be scavengers as well. The question is: If the order includes one mixed, two predatory, two scavenger families could we still safely assume that the ancestral state is predatory?

While is still in debate if new world vultures (Cathartidae) are a family part of Accipitriformes or a separated order, it is clear the they are the sister group to all other Accipitriformes, including old world vultures, which are well nested within other Accipitriformes. While it is possible that the

ancestor of all Accipitriformes including New-world vultures, was a scavenger, the two independently evolved Old-world vultures clades are well nested within fully predatory clades, and the ancestral state reconstruction of diet for all Accipitriformes (which we did not show) confirms that for both Old-world vultures clades there was independent transitions from vertivore to scavenger. So we think that it is safe to assume that at least for the Old-world vultures, the ancestral state was predatory.

8. Lines 186-189 The way parrots grasp object is not incorrectly described but I found it a bit misleading in the text and in the Figure. It is true that the foot is being turned inward in those birds relative to their ankle bones. I feel when we think about turning a limb inward while grasping we (being primates) tend to intuitively think about the orientation of the palm. As in most parrot the palm is oriented outwards (but inwards in Amazon and Eclectus) while grasping I found this confusing.

We understand the concern from the reviewer but we think that our description is anatomically correct and also accompanied with multiple pictures in figure 4 which clearly show what is described.

9. Line 233 I would recommend replacing the word 'confirm' with 'support'.

We have change confirm to support.

10. Lines 249-252 Shrikes and drongos also feed on small vertebrates and eat them up in the trees. Could this combination foster lifting behavior? Note perhaps also that many insectivorous birds swallow their prey in whole and do not have to process it first.

Small vertebrates are only a small percentage of the diet of both of these groups and almost all species are considered invertivores, this is, more than 60 % of their diets are invertebrates. Therefore, we don't think that this is a important trait in the evolution of lifting behavior. In fact, in most cases in this clades, the prey being lifted is an insect and not a vertebrate.

11. Lines 269-282 It would be nice to provide some alternative explanations. Notably, parrots also have other traits (than possibly having meat-eating ancestors) that may have fostered foot use. Their zygodactyl feet are probably an adaption to clambering and climbing in the canopy, but they could arguably be better for grasping than anisodactyl feet. Parrots dehusk complex and often large seeds with their beak. This can be a tedious process in which the seed has to be turned in between manipulation bouts. With larger seeds this is not always possible to keep them in the beak while turning them. Note also that many parrots do opportunistically feed on insects and small vertebrates.

There reviewer is correct that other possibilities exist. We have added a small paragraph at the end of that sections that reads “

“Nevertheless, is possible that the ability to grasp and bring objects to the beak as evolved independently, and for different reasons in different groups of birds. In the case of parrots for example, is possible that the combination of a zygodactyl foot, which allows for a firm grip(Ksepka and Clarke 2012), and an diet based on extractive foraging of seeds and fruits(Forshaw and Knight 2010) it is what drove the evolution of the this behavior. “

12. Line 330 Phrases like or exactly those phrases?

We look exactly those phrases. To reflect this, we have amended the text and now reads:

“...journals in the Biodiversity Heritage Library (BHL) for the phrases “a foot,” “under a foot,” “its foot,” “its feet,” “held under a foot,” and others”

13. Lines 452-454 You might mention that still little is known about the details of the wildlife ecology of many avian species and that some niches may thus not be up to date.

The reviewer is right that diet and niche data remains poorly known for many species. We have added a sentence to reflect this adn now reads:

“Diet information for all species was obtained from Pigot et al 31, which used an updated version of the EltonTraits dataset69,70. While updated, the ecology and diet of many birds species remains poorly known and therefore some of this data is inevitably inaccurate and may change in the future.”

14. Table 1: This needs some clarifications. What is the difference between ‘use of toes’ and ‘grasping’? what is the difference between ‘use of one leg and free grasping (without flight)’?

We have amended Table 1 for clarity. In the case of “use of toes” we counted this behavior when object was held with only some toes and not the whole foot/claw. To reflect this, it now reads: “Object is held or grasped with only some toes”

In the case of “use of one leg” this behavior was counted when the object was manipulated with only one (as oppose to booth at the same time). To better reflect this, it now reads “Only one leg is used to hold or grasp”

Other changes:

Since the publication of our preprint, we have been directed by another researcher to several additional images and videos of birds using their feet, particularly raptors. Because of this we have updated figure 3 and the two supplementary datasets. We also amended line 159-160 to reflect the new data, which shows that the ability to bring the foot to the beak in Accipitriformes has evolved in some additional genera. Figure 5 was also slight amended to add additional behavior in one species of drongo.

REVIEWERS' COMMENTS:

Reviewer #2 (Remarks to the Author):

I feel like the authors did a good job in adjusting their manuscript according to previous suggestions or provided sufficient explanation when they did not. I think that this is a very nice manuscript with a great fit for this journal.

Just one minor issue: I still feel like the word skill should only be used in a context where a trait is actively acquired (as most dictionaries would corroborate with that).

The fact that a word has been used wrongly in the past is no justification to continue to do so.